# Sequential Attend, Infer, Repeat: Generative Modelling of Moving Objects

**Adam R. Kosiorek**[*][§][†]  **Hyunjik Kim**[†]  **Ingmar Posner**[§]  **Yee Whye Teh**[†]

[§] **Applied Artificial Intelligence Lab**
Oxford Robotics Institute
University of Oxford

[†] **Department of Statistics**
University of Oxford

## Abstract

We present Sequential Attend, Infer, Repeat (SQAIR), an interpretable deep generative model for videos of moving objects. It can reliably discover and track objects throughout the sequence of frames, and can also generate future frames conditioning on the current frame, thereby simulating expected motion of objects. This is achieved by explicitly encoding object presence, locations and appearances in the latent variables of the model. SQAIR retains all strengths of its predecessor, Attend, Infer, Repeat (AIR, Eslami et al., 2016), including learning in an unsupervised manner, and addresses its shortcomings. We use a moving multi-MNIST dataset to show limitations of AIR in detecting overlapping or partially occluded objects, and show how SQAIR overcomes them by leveraging temporal consistency of objects. Finally, we also apply SQAIR to real-world pedestrian CCTV data, where it learns to reliably detect, track and generate walking pedestrians with no supervision.

## 1   Introduction

The ability to identify objects in their environments and to understand relations between them is a cornerstone of human intelligence (Kemp and Tenenbaum, 2008). Arguably, in doing so we rely on a notion of spatial and temporal consistency which gives rise to an expectation that objects do not appear out of thin air, nor do they spontaneously vanish, and that they can be described by properties such as location, appearance and some dynamic behaviour that explains their evolution over time. We argue that this notion of consistency can be seen as an *inductive bias* that improves the efficiency of our learning. Equally, we posit that introducing such a bias towards spatio-temporal consistency into our models should greatly reduce the amount of supervision required for learning.

One way of achieving such inductive biases is through model structure. While recent successes in deep learning demonstrate that progress is possible without explicitly imbuing models with interpretable structure (LeCun, Bengio, et al., 2015), recent works show that introducing such structure into deep models can indeed lead to favourable inductive biases improving performance e.g. in convolutional networks (LeCun, Boser, et al., 1989) or in tasks requiring relational reasoning (Santoro et al., 2017). Structure can also make neural networks useful in new contexts by significantly improving generalization, data efficiency (Jacobsen et al., 2016) or extending their capabilities to unstructured inputs (Graves et al., 2016).

Attend, Infer, Repeat (AIR), introduced by Eslami et al., 2016, is a notable example of such a structured probabilistic model that relies on deep learning and admits efficient amortized inference. Trained without any supervision, AIR is able to decompose a visual scene into its constituent components and to generate a (learned) number of latent variables that explicitly encode the location and appearance of each object. While this approach is inspiring, its focus on modelling individual (and thereby inherently static) scenes leads to a number of limitations. For example, it often merges two objects that are close together into one since no temporal context is available to distinguish between them.

---

[*]Corresponding author: adamk@robots.ox.ac.uk

Similarly, we demonstrate that AIR struggles to identify partially occluded objects, e.g. when they extend beyond the boundaries of the scene frame (see Figure 7 in Section 4.1).

Our contribution is to mitigate the shortcomings of AIR by introducing a sequential version that models sequences of frames, enabling it to discover and track objects over time as well as to generate convincing extrapolations of frames into the future. We achieve this by leveraging temporal information to learn a richer, more capable generative model. Specifically, we extend AIR into a spatio-temporal state-space model and train it on unlabelled image sequences of dynamic objects. We show that the resulting model, which we name Sequential AIR (SQAIR), retains the strengths of the original AIR formulation while outperforming it on moving MNIST digits.

The rest of this work is organised as follows. In Section 2, we describe the generative model and inference of AIR. In Section 3, we discuss its limitations and how it can be improved, thereby introducing Sequential Attend, Infer, Repeat (SQAIR), our extension of AIR to image sequences. In Section 4, we demonstrate the model on a dataset of multiple moving MNIST digits (Section 4.1) and compare it against AIR trained on each frame and Variational Recurrent Neural Network (VRNN) of Chung et al., 2015 with convolutional architectures, and show the superior performance of SQAIR in terms of log marginal likelihood and interpretability of latent variables. We also investigate the utility of inferred latent variables of SQAIR in downstream tasks. In Section 4.2 we apply SQAIR on real-world pedestrian CCTV data, where SQAIR learns to reliably detect, track and generate walking pedestrians without any supervision. Code for the implementation on the MNIST dataset[2] and the results video[3] are available online.

## 2   Attend, Infer, Repeat (AIR)

AIR, introduced by Eslami et al., 2016, is a structured variational auto-encoder (VAE) capable of decomposing a static scene $\mathbf{x}$ into its constituent objects, where each object is represented as a separate triplet of continuous latent variables $\mathbf{z} = \{\mathbf{z}^{\text{what},i}, \mathbf{z}^{\text{where},i}, z^{\text{pres},i}\}_{i=1}^{n}$, $n \in \mathbb{N}$ being the (random) number of objects in the scene. Each triplet of latent variables explicitly encodes position, appearance and presence of the respective object, and the model is able to infer the number of objects present in the scene. Hence it is able to count, locate and describe objects in the scene, all learnt in an unsupervised manner, made possible by the inductive bias introduced by the model structure.

**Generative Model** The generative model of AIR is defined as follows

$$p_\theta(n) = \text{Geom}(n \mid \theta), \qquad p_\theta(\mathbf{z}^{\text{w}} \mid n) = \prod_{i=1}^{n} p_\theta(\mathbf{z}^{w,i}) = \prod_{i=1}^{n} \mathcal{N}(\mathbf{z}^{w,i} \mid \mathbf{0}, \mathbf{I}),$$

$$p_\theta(\mathbf{x} \mid \mathbf{z}) = \mathcal{N}(\mathbf{x} \mid \mathbf{y}_t, \sigma_x^2 \boldsymbol{I}), \qquad \text{with } \mathbf{y}_t = \sum_{i=1}^{n} \text{h}_\theta^{\text{dec}}(\mathbf{z}^{\text{what},i}, \mathbf{z}^{\text{where},i}), \tag{1}$$

where $\mathbf{z}^{\text{w},i} \coloneqq (\mathbf{z}^{\text{what},i}, \mathbf{z}^{\text{where},i})$, $z^{\text{pres},i} = 1$ for $i = 1 \dots n$ and $h_\theta^{\text{dec}}$ is the object decoder with parameters $\theta$. It is composed of a *glimpse decoder* $f_\theta^{\text{dec}} : \mathbf{g}_t^i \mapsto \mathbf{y}_t^i$, which constructs an image patch and a spatial transformer (ST, Jaderberg et al., 2015), which scales and shifts it according to $\mathbf{z}^{\text{where}}$; see Figure 1 for details.

**Inference** Eslami et al., 2016 use a sequential inference algorithm, where latent variables are inferred one at a time; see Figure 2. The number of inference steps $n$ is given by $z^{\text{pres},1:n+1}$, a random vector of $n$ ones followed by a zero. The $\mathbf{z}^i$ are sampled sequentially from

$$q_\phi(\mathbf{z} \mid \mathbf{x}) = q_\phi(z^{\text{pres},n+1} = 0 \mid \mathbf{z}^{\text{w},1:n}, \mathbf{x}) \prod_{i=1}^{n} q_\phi(\mathbf{z}^{\text{w},i}, z^{\text{pres},i} = 1 \mid \mathbf{z}^{1:i-1}, \mathbf{x}), \tag{2}$$

where $q_\phi$ is implemented as a neural network with parameters $\phi$. To implement explaining away, e.g. to avoid encoding the same object twice, it is vital to capture the dependency of $\mathbf{z}^{\text{w},i}$ and $z^{\text{pres},i}$ on $\mathbf{z}^{1:i-1}$ and $\mathbf{x}$. This is done using a recurrent neural network (RNN) $R_\phi$ with hidden state $\boldsymbol{h}^i$, namely: $\boldsymbol{\omega}^i, \boldsymbol{h}^i = R_\phi(\mathbf{x}, \mathbf{z}^{i-1}, \boldsymbol{h}^{i-1})$. The outputs $\boldsymbol{\omega}^i$, which are computed iteratively and depend on the previous latent variables (*cf.* Algorithm 3), parametrise $q_\phi(\mathbf{z}^{\text{w},i}, z^{\text{pres},i} \mid \mathbf{z}^{1:i-1}, \mathbf{x})$. For simplicity the latter is assumed to factorise such that $q_\phi(\mathbf{z}^{\text{w}}, \mathbf{z}^{\text{pres}} \mid \mathbf{z}^{1:i-1}, \mathbf{x}) = q_\phi(z^{\text{pres},n+1} = 0 \mid \boldsymbol{\omega}^{n+1}) \prod_{i=1}^{n} q_\phi(\mathbf{z}^{\text{w},i} \mid \boldsymbol{\omega}^i) q_\phi(z^{\text{pres},i} = 1 \mid \boldsymbol{\omega}^i)$.

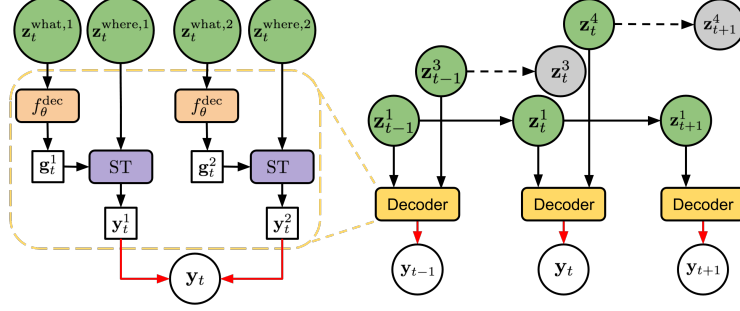

Figure 1: *Left*: Generation in AIR. The image mean $\mathbf{y}_t$ is generated by first using the *glimpse decoder* $f_\theta^{\text{dec}}$ to map the *what* variables into glimpses $\mathbf{g}_t$, transforming them with the *spatial transformer* ST according to the *where* variables and summing up the results. *Right*: Generation in SQAIR. When new objects enter the frame, new latent variables (here, $\mathbf{z}_t^4$) are sampled from the *discovery* prior. The temporal evolution of already present objects is governed by the *propagation* prior, which can choose to forget some variables (here, $\mathbf{z}_t^3$ and $\mathbf{z}_{t+1}^4$) when the object moves out of the frame. The image generation process, which mimics the left-hand side of the figure, is abstracted in the *decoder* block.

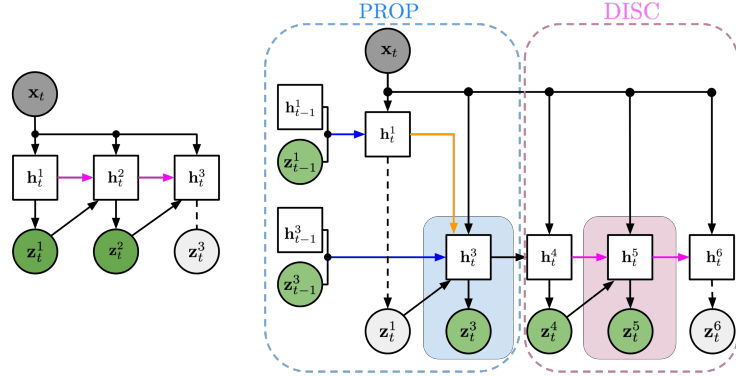

Figure 2: *Left*: Inference in AIR. The pink RNN attends to the image sequentially and produces one latent variable $\mathbf{z}_t^i$ at a time. Here, it decides that two latent variables are enough to explain the image and $\mathbf{z}_t^3$ is not generated. *Right*: Inference in SQAIR starts with the Propagation (PROP) phase. PROP iterates over latent variables from the previous time-step $t-1$ and updates them based on the new observation $\mathbf{x}_t$. The blue RNN runs forward in time to update the hidden state of each object, to model its change in appearance and location throughout time. The orange RNN runs across all current objects and models the relations between different objects. Here, when attending to $\mathbf{z}_{t-1}^1$, it decides that the corresponding object has disappeared from the frame and *forgets* it. Next, the Discovery (DISC) phase detects new objects as in AIR, but in SQAIR it is also conditioned on the results of PROP, to prevent rediscovering objects. See Figure 3 for details of the colored RNNs.

## 3   Sequential Attend-Infer-Repeat

While capable of decomposing a scene into objects, AIR only describes single images. Should we want a similar decomposition of an image sequence, it would be desirable to do so in a temporally consistent manner. For example, we might want to detect objects of the scene as well as infer dynamics and track identities of any persistent objects. Thus, we introduce Sequential Attend, Infer, Repeat (SQAIR), whereby AIR is augmented with a state-space model (SSM) to achieve temporal consistency in the generated images of the sequence. The resulting probabilistic model is composed of two parts: Discovery (DISC), which is responsible for detecting (or introducing, in the case of the generation) new objects at every time-step (essentially equivalent to AIR), and Propagation (PROP), responsible for updating (or forgetting) latent variables from the previous time-step given the new observation (image), effectively implementing the temporal SSM. We now formally introduce SQAIR by first describing its generative model and then the inference network.

**Generative Model** The model assumes that at every-time step, objects are first propagated from the previous time-step (PROP). Then, new objects are introduced (DISC). Let $t \in \mathbb{N}$ be the current time-step. Let $\mathcal{P}_t$ be the set of objects propagated from the previous time-step and let $\mathcal{D}_t$ be the set of objects discovered at the current time-step, and let $\mathcal{O}_t = \mathcal{P}_t \cup \mathcal{D}_t$ be the set of all objects present at time-step $t$. Consequently, at every time step, the model retains a set of latent variables $\mathbf{z}_t^{\mathcal{P}_t} = \{\mathbf{z}_t^i\}_{i \in \mathcal{P}_t}$, and

generates a set of new latent variables $\mathbf{z}_t^{\mathcal{D}_t} = \{\mathbf{z}_t^i\}_{i \in \mathcal{D}_t}$. Together they form $\mathbf{z}_t := [\mathbf{z}_t^{\mathcal{P}_t}, \mathbf{z}_t^{\mathcal{D}_t}]$, where the representation of the $i^{\text{th}}$ object $\mathbf{z}_t^i := [\mathbf{z}_t^{\text{what},i}, \mathbf{z}_t^{\text{where},i}, z_t^{\text{pres},i}]$ is composed of three components (as in AIR): $\mathbf{z}_t^{\text{what},i}$ and $\mathbf{z}_t^{\text{where},i}$ are real vector-valued variables representing appearance and location of the object, respectively. $z_t^{\text{pres},i}$ is a binary variable representing whether the object is present at the given time-step or not.

At the first time-step ($t = 1$) there are no objects to propagate, so we sample $D_1$, the number of objects at $t = 1$, from the discovery prior $p^D(D_1)$. Then for each object $i \in \mathcal{D}_t$, we sample latent variables $\mathbf{z}_t^{\text{what},i}, \mathbf{z}_t^{\text{where},i}$ from $p^D(z_1^i \mid D_1)$. At time $t = 2$, the *propagation* step models which objects from $t = 1$ are propagated to $t = 2$, and which objects disappear from the frame, using the binary random variable $(z_t^{\text{pres},i})_{i \in \mathcal{P}_t}$. The *discovery* step at $t = 2$ models new objects that enter the frame, with a similar procedure to $t = 1$: sample $D_2$ (which depends on $\mathbf{z}_2^{\mathcal{P}_2}$) then sample $(\mathbf{z}_2^{\text{what},i}, \mathbf{z}_2^{\text{where},i})_{i \in \mathcal{D}_2}$. This procedure of propagation and discovery recurs for $t = 2, \ldots T$. Once the $\mathbf{z}_t$ have been formed, we may generate images $\mathbf{x}_t$ using the exact same generative distribution $p_\theta(\mathbf{x}_t \mid \mathbf{z}_t)$ as in AIR (*cf.* Equation (1), Fig. 1, and Algorithm 1). In full, the generative model is:

$$p(\mathbf{x}_{1:T}, \mathbf{z}_{1:T}, D_{1:T}) = p^D(D_1, \mathbf{z}_1^{\mathcal{D}_1}) \prod_{t=2}^{T} p^D(D_t, \mathbf{z}_t^{\mathcal{D}_t} | \mathbf{z}_t^{\mathcal{P}_t}) p^P(\mathbf{z}_t^{\mathcal{P}_t} | \mathbf{z}_{t-1}) p_\theta(\mathbf{x}_t | \mathbf{z}_t), \qquad (3)$$

The *discovery prior* $p^D(D_t, \mathbf{z}_t^{\mathcal{D}_t} | \mathbf{z}_t^{\mathcal{P}_t})$ samples latent variables for new objects that enter the frame. The *propagation prior* $p^P(\mathbf{z}_t^{\mathcal{P}_t} | \mathbf{z}_{t-1})$ samples latent variables for objects that persist in the frame and removes latents of objects that disappear from the frame, thereby modelling dynamics and appearance changes. Both priors are learned during training. The exact forms of the priors are given in Appendix B.

**Inference** Similarly to AIR, inference in SQAIR can capture the number of objects and the representation describing the location and appearance of each object that is necessary to explain every image in a sequence. As with generation, inference is divided into PROP and DISC. During PROP, the inference network achieves two tasks. Firstly, the latent variables from the previous time step are used to infer the current ones, modelling the change in location and appearance of the corresponding objects, thereby attaining temporal consistency. This is implemented by the *temporal* RNN $\mathrm{R}_\phi^T$, with hidden states $\boldsymbol{h}_t^T$ (recurs in $t$). Crucially, it does not access the current image directly, but uses the output of the *relation* RNN (*cf.* Santoro et al., 2017). The relation RNN takes relations between objects into account, thereby implementing the *explaining away* phenomenon; it is essential for capturing any interactions between objects as well as occlusion (or overlap, if one object is occluded by another). See Figure 7 for an example. These two RNNs together decide whether to retain or to forget objects that have been propagated from the previous time step. During DISC, the network infers further latent variables that are needed to describe any new objects that have entered the frame. All latent variables remaining after PROP and DISC are passed on to the next time step.

See Figures 2 and 3 for the inference network structure . The full variational posterior is defined as

$$q_\phi(D_{1:t}, \mathbf{z}_{1:T} \mid \mathbf{x}_{1:T}) = \prod_{t=1}^{T} q_\phi^D\left(D_t, \mathbf{z}_t^{\mathcal{D}_t} \mid \mathbf{x}_t, \mathbf{z}_t^{\mathcal{P}_t}\right) \prod_{i \in \mathcal{O}_{t-1}} q_\phi^P\left(\mathbf{z}_t^i \mid \mathbf{z}_{t-1}^i, \boldsymbol{h}_t^{T,i}, \boldsymbol{h}_t^{R,i}\right). \qquad (4)$$

Discovery, described by $q_\phi^D$, is very similar to the full posterior of AIR, *cf.* Equation (2). The only difference is the conditioning on $\mathbf{z}_t^{\mathcal{P}_t}$, which allows for a different number of discovered objects at each time-step and also for objects explained by PROP not to be explained again. The second term, or $q_\phi^P$, describes propagation. The detailed structures of $q_\phi^D$ and $q_\phi^P$ are shown in Figure 3, while all the pertinent algorithms and equations can be found in Appendices A and C, respectively.

**Learning** We train SQAIR as an importance-weighted auto-encoder (IWAE) of Burda et al., 2016. Specifically, we maximise the importance-weighted evidence lower-bound $\mathcal{L}_{\text{IWAE}}$, namely

$$\mathcal{L}_{\text{IWAE}} = \mathbb{E}_{\mathbf{x}_{1:T} \sim p_{\text{data}}(\mathbf{x}_{1:T})}\left[ \mathbb{E}_q\left[ \log \frac{1}{K} \sum_{k=1}^{K} \frac{p_\theta(\mathbf{x}_{1:T}, \mathbf{z}_{1:T})}{q_\phi(\mathbf{z}_{1:T} \mid \mathbf{x}_{1:T})} \right] \right]. \qquad (5)$$

To optimise the above, we use RMSPROP, $K = 5$ and batch size of 32. We use the VIMCO gradient estimator of Mnih and Rezende, 2016 to backpropagate through the discrete latent variables $z^{\text{pres}}$,

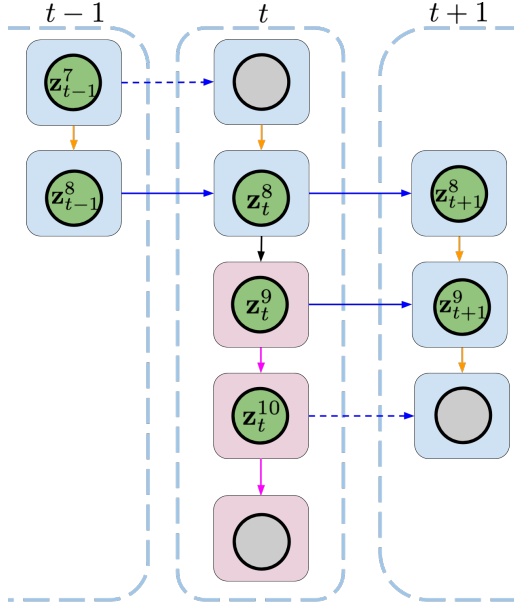

Figure 3: *Left*: Interaction between PROP and DISC in SQAIR. Firstly, objects are propagated to time $t$, and object $i = 7$ is dropped. Secondly, DISC tries to discover new objects. Here, it manages to find two objects: $i = 9$ and $i = 10$. The process recurs for all remaining time-steps. Blue arrows update the temporal hidden state, orange ones infer relations between objects, pink ones correspond to discovery. *Bottom*: Information flow in a single discovery block (*left*) and propagation block (*right*). In DISC we first predict *where* and extract a glimpse. We then predict *what* and *presence*. PROP starts with extracting a glimpse at a candidate location and updating *where*. Then it follows a procedure similar to DISC, but takes the respective latent variables from the previous time-step into account. It is approximately two times more computationally expensive than DISC. For details, see Algorithms 2 and 3 in Appendix A.

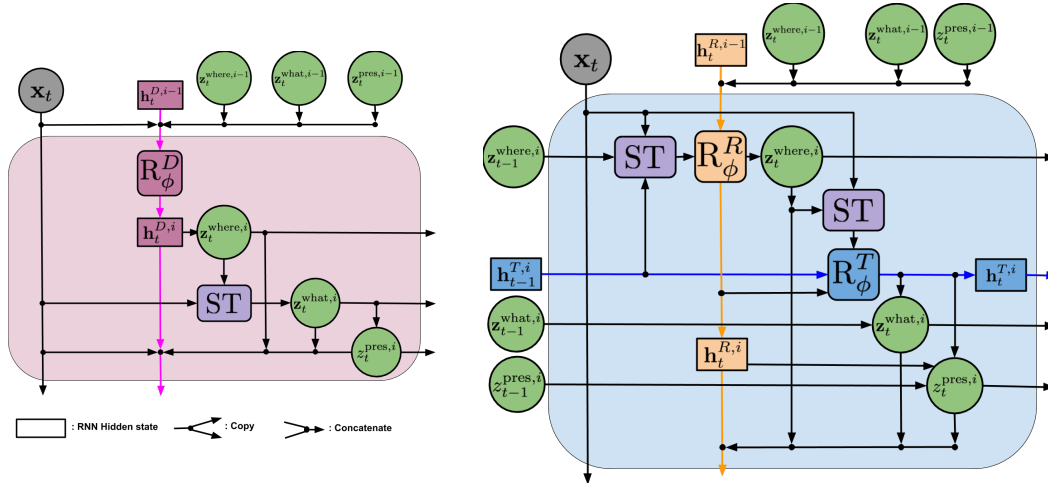

and use reparameterisation for the continuous ones (Kingma and Welling, 2013). We also tried to use NVIL of Mnih and Gregor, 2014 as in the original work on AIR, but found it very sensitive to hyper-parameters, fragile and generally under-performing.

## 4 Experiments

We evaluate SQAIR on two datasets. Firstly, we perform an extensive evaluation on moving MNIST digits, where we show that it can learn to reliably detect, track and generate moving digits (Section 4.1). Moreover, we show that SQAIR can simulate moving objects into the future — an outcome it has not been trained for. We also study the utility of learned representations for a downstream task. Secondly, we apply SQAIR to real-world pedestrian CCTV data from static cameras (*DukeMTMC*, Ristani et al., 2016), where we perform background subtraction as pre-processing. In this experiment, we show that SQAIR learns to detect, track, predict and generate walking pedestrians without human supervision.

### 4.1 Moving multi-MNIST

The dataset consists of sequences of length 10 of multiple moving MNIST digits. All images are of size $50 \times 50$ and there are zero, one or two digits in every frame (with equal probability). Sequences are generated such that no objects overlap in the first frame, and all objects are present through the sequence; the digits can move out of the frame, but always come back. See Appendix F for an experiment on a harder version of this dataset. There are 60,000 training and 10,000 testing sequences created from the respective MNIST datasets. We train two variants of SQAIR: the MLP-SQAIR uses

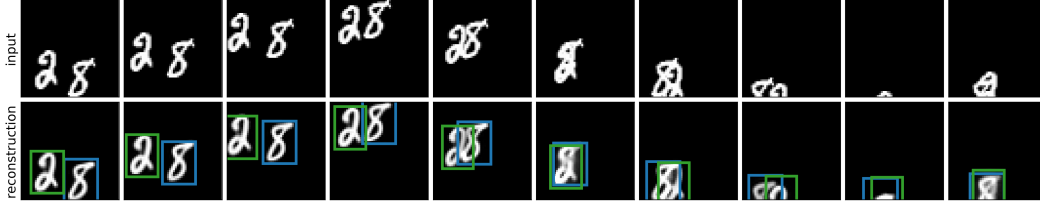

Figure 4: Input images (top) and SQAIR reconstructions with marked glimpse locations (bottom). For more examples, see Figure 13 in Appendix H.

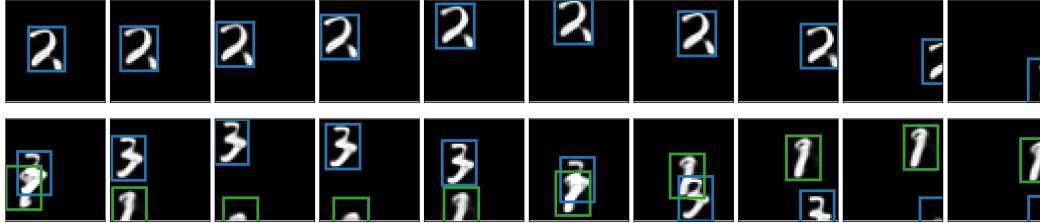

Figure 5: Samples from SQAIR. Both motion and appearance are consistent through time, thanks to the propagation part of the model. For more examples, see Figure 15 in Appendix H.

only fully-connected networks, while the CONV-SQAIR replaces the networks used to encode images and glimpses with convolutional ones; it also uses a subpixel-convolution network as the glimpse decoder (Shi et al., 2016). See Appendix D for details of the model architectures and the training procedure.

We use AIR and VRNN (Chung et al., 2015) as baselines for comparison. VRNN can be thought of as a sequential VAE with an RNN as its deterministic backbone. Being similar to a VAE, its latent variables are not structured, nor easily interpretable. For a fair comparison, we control the latent dimensionality of VRNN and the number of learnable parameters. We provide implementation details in Appendix D.3.

The quantitative analysis consists of comparing all models in terms of the marginal log-likelihood $\log p_\theta(\mathbf{x}_{1:T})$ evaluated as the $\mathcal{L}_{\text{IWAE}}$ bound with $K = 1000$ particles, reconstruction quality evaluated as a single-sample approximation of $\mathbb{E}_{q_\phi}[\log p_\theta(\mathbf{x}_{1:T} \mid \mathbf{z}_{1:T})]$ and the KL-divergence between the approximate posterior and the prior (Table 1). Additionally, we measure the accuracy of the number of objects modelled by SQAIR and AIR. SQAIR achieves superior performance across a range of metrics — its convolutional variant outperforms both AIR and the corresponding VRNN in terms of model evidence and reconstruction performance. The KL divergence for SQAIR is almost twice as low as for VRNN and by a yet larger factor for AIR. We can interpret KL values as an indicator of the ability to compress, and we can treat SQAIR/AIR type of scheme as a version of run-length encoding. While VRNN has to use information to explicitly describe every part of the image, even if some parts are empty, SQAIR can explicitly allocate content information ($\mathbf{z}^{\text{what}}$) to specific parts of the image (indicated by $\mathbf{z}^{\text{where}}$). AIR exhibits the highest values of KL, but this is due to encoding every frame of the sequence independently — its prior cannot take *what* and *where* at the previous time-step into account, hence higher KL. The fifth column of Table 1 details the object counting accuracy, that is indicative of the quality of the approximate posterior. It is measured as the sum of $z_t^{\text{pres}}$ for a given frame against the true number of objects in that frame. As there is no $z^{\text{pres}}$ for VRNN no score is provided. Perhaps surprisingly, this metric is much higher for SQAIR than for AIR. This is because AIR mistakenly infers overlapping objects as a single object. Since SQAIR can incorporate temporal

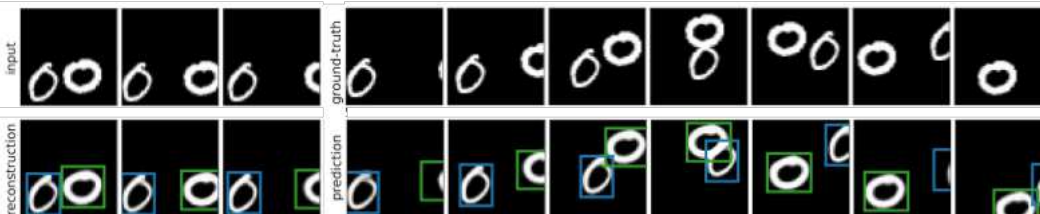

Figure 6: The first three frames are input to SQAIR, which generated the rest conditional on the first frames.

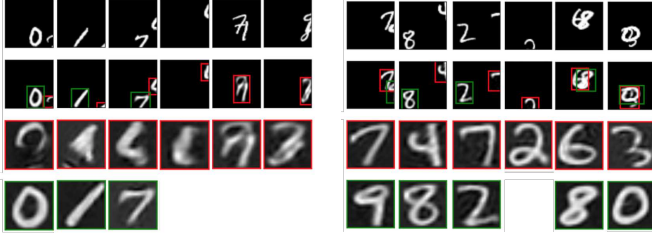

Figure 7: Inputs, reconstructions with marked glimpse locations and reconstructed glimpses for AIR (left) and SQAIR (right). SQAIR can model partially visible and heavily overlapping objects by aggregating temporal information.

| | $\log p_\theta(\mathbf{x}_{1:T})$ | $\log p_\theta(\mathbf{x}_{1:T} \mid \mathbf{z}_{1:T})$ | $\mathrm{KL}(q_\phi \parallel p_\theta)$ | Counting | Addition |
|---|---|---|---|---|---|
| CONV-SQAIR | **6784.8** | **6923.8** | **134.6** | 0.9974 | 0.9990 |
| MLP-SQAIR | 6617.6 | 6786.5 | 164.5 | **0.9986** | **0.9998** |
| MLP-AIR | 6443.6 | 6830.6 | 352.6 | 0.9058 | 0.8644 |
| CONV-VRNN | 6561.9 | 6737.8 | 270.2 | n/a | 0.8536 |
| MLP-VRNN | 5959.3 | 6108.7 | 218.3 | n/a | 0.8059 |

Table 1: SQAIR achieves higher performance than the baselines across a range of metrics. The third column refers to the Kullback-Leibler (KL) divergence between the approximate posterior and the prior. Counting refers to accuracy of the inferred number of objects present in the scene, while addition stands for the accuracy of a supervised digit addition experiment, where a classifier is trained on the learned latent representations of each frame.

information, it does not exhibit this failure mode (*cf.* Figure 7). Next, we gauge the utility of the learnt representations by using them to determine the sum of the digits present in the image (Table 1, column six). To do so, we train a 19-way classifier (mapping from any combination of up to two digits in the range $[0, 9]$ to their sum) on the extracted representations and use the summed labels of digits present in the frame as the target. Appendix D contains details of the experiment. SQAIR significantly outperforms AIR and both variants of VRNN on this tasks. VRNN under-performs due to the inability of disentangling overlapping objects, while both VRNN and AIR suffer from low temporal consistency of learned representations, see Appendix H. Finally, we evaluate SQAIR qualitatively by analyzing reconstructions and samples produced by the model against reconstructions and samples from VRNN. We observe that samples and reconstructions from SQAIR are of better quality and, unlike VRNN, preserve motion and appearance consistently through time. See Appendix H for direct comparison and additional examples. Furthermore, we examine conditional generation, where we look at samples from the generative model of SQAIR conditioned on three images from a real sequence (see Figure 6). We see that the model can preserve appearance over time, and that the simulated objects follow similar trajectories, which hints at good learning of the motion model (see Appendix H for more examples). Figure 7 shows reconstructions and corresponding glimpses of AIR and SQAIR. Unlike SQAIR, AIR is unable to recognize objects from partial observations, nor can it distinguish strongly overlapping objects (it treats them as a single object; columns five and six in the figure). We analyze failure cases of SQAIR in Appendix G.

## 4.2 Generative Modelling of Walking Pedestrians

To evaluate the model in a more challenging, real-world setting, we turn to data from static CCTV cameras of the *DukeMTMC* dataset (Ristani et al., 2016). As part of pre-precessing, we use standard background subtraction algorithms (Itseez, 2015). In this experiment, we use 3150 training and 350 validation sequences of length 5. For details of model architectures, training and data pre-processing, see Appendix E. We evaluate the model qualitatively by examining reconstructions, conditional samples (conditioned on the first four frames) and samples from the prior (Figure 8 and Appendix I). We see that the model learns to reliably detect and track walking pedestrians, even when they are close to each other.

There are some spurious detections and re-detections of the same objects, which is mostly caused by imperfections of the background subtraction pipeline — backgrounds are often noisy and there are sudden appearance changes when a part of a person is treated as background in the pre-processing pipeline. The object counting accuracy in this experiment is 0.5712 on the validation dataset, and we noticed that it does increase with the size of the training set. We also had to use early stopping to prevent overfitting, and the model was trained for only 315k iterations ($> 1$M for MNIST experiments). Hence, we conjecture that accuracy and marginal likelihood can be further improved by using a bigger dataset.

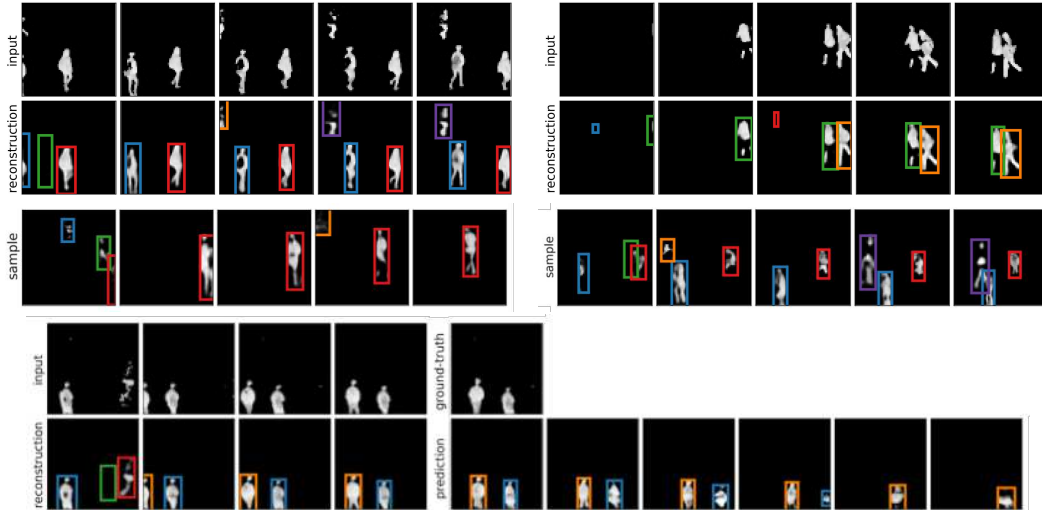

Figure 8: Inputs on the top, reconstructions in the second row, samples in the third row; rows four and five contain inputs and conditional generation: the first four frames in the last row are reconstructions, while the remaining ones are predicted by sampling from the prior. There is no ground-truth, since we used sequences of length five of training and validation.

## 5 Related Work

**Object Tracking** There have been many approaches to modelling objects in images and videos. Object detection and tracking are typically learned in a supervised manner, where object bounding boxes and often additional labels are part of the training data. Single-object tracking commonly use Siamese networks, which can be seen as an RNN unrolled over two time-steps (Valmadre et al., 2017). Recently, Kosiorek et al., 2017 used an RNN with an attention mechanism in the HART model to predict bounding boxes for single objects, while robustly modelling their motion and appearance. Multi-object tracking is typically attained by detecting objects and performing data association on bounding-boxes (Bewley et al., 2016). Schulter et al., 2017 used an end-to-end supervised approach that detects objects and performs data association. In the unsupervised setting, where the training data consists of only images or videos, the dominant approach is to distill the inductive bias of spatial consistency into a discriminative model. Cho et al., 2015 detect single objects and their parts in images, and Kwak et al., 2015; Xiao and Jae Lee, 2016 incorporate temporal consistency to better track single objects. SQAIR is unsupervised and hence it does not rely on bounding boxes nor additional labels for training, while being able to learn arbitrary motion and appearance models similarly to HART (Kosiorek et al., 2017). At the same time, is inherently multi-object and performs data association implicitly (*cf.* Appendix A). Unlike the other unsupervised approaches, temporal consistency is baked into the model structure of SQAIR and further enforced by lower KL divergence when an object is tracked.

**Video Prediction** Many works on video prediction learn a deterministic model conditioned on the current frame to predict the future ones (Ranzato et al., 2014; Srivastava et al., 2015). Since these models do not model uncertainty in the prediction, they can suffer from the multiple futures problem — since perfect prediction is impossible, the model produces blurry predictions which are a mean of possible outcomes. This is addressed in stochastic latent variable models trained using variational inference to generate multiple plausible videos given a sequence of images (Babaeizadeh et al., 2017; Denton and Fergus, 2018). Unlike SQAIR, these approaches do not model objects or their positions explicitly, thus the representations they learn are of limited interpretability.

**Learning Decomposed Representations of Images and Videos** Learning decomposed representations of object appearance and position lies at the heart of our model. This problem can be also seen as perceptual grouping, which involves modelling pixels as spatial mixtures of entities. Greff, Rasmus, et al., 2016 and Greff, Steenkiste, et al., 2017 learn to decompose images into separate entities by iterative refinement of spatial clusters using either learned updates or the Expectation Maximization algorithm; Ilin et al., 2017 and Steenkiste et al., 2018 extend these approaches to videos, achieving very similar results to SQAIR. Perhaps the most similar work to ours is the concurrently developed model of Hsieh et al., 2018. The above approaches rely on iterative inference procedures,

but do not exhibit the object-counting behaviour of SQAIR. For this reason, their computational complexities are proportional to the predefined maximum number of objects, while SQAIR can be more computationally efficient by adapting to the number of objects currently present in an image.

Another interesting line of work is the GAN-based unsupervised video generation that decomposes motion and content (Tulyakov et al., 2018; Denton and Birodkar, 2017). These methods learn interpretable features of content and motion, but deal only with single objects and do not explicitly model their locations. Nonetheless, adversarial approaches to learning structured probabilistic models of objects offer a plausible alternative direction of research.

**Bayesian Nonparametric Models**  To the best of our knowledge, Neiswanger and Wood, 2012 is the only known approach that models pixels belonging to a variable number of objects in a video together with their locations in the generative sense. This work uses a Bayesian nonparametric (BNP) model, which relies on mixtures of Dirichlet processes to cluster pixels belonging to an object. However, the choice of the model necessitates complex inference algorithms involving Gibbs sampling and Sequential Monte Carlo, to the extent that any sensible approximation of the marginal likelihood is infeasible. It also uses a fixed likelihood function, while ours is learnable.

The object appearance-persistence-disappearance model in SQAIR is reminiscent of the Markov Indian buffet process (MIBP) of Gael et al., 2009, another BNP model. MIBP was used as a model for blind source separation, where multiple sources contribute toward an audio signal, and can appear, persist, disappear and reappear independently. The prior in SQAIR is similar, but the crucial differences are that SQAIR combines the BNP prior with flexible neural network models for the dynamics and likelihood, as well as variational learning via amortized inference. The interface between deep learning and BNP, and graphical models in general, remains a fertile area of research.

## 6   Discussion

In this paper we proposed SQAIR, a probabilistic model that extends AIR to image sequences, and thereby achieves temporally consistent reconstructions and samples. In doing so, we enhanced AIR's capability of disentangling overlapping objects and identifying partially observed objects.

This work continues the thread of Greff, Steenkiste, et al., 2017, Steenkiste et al., 2018 and, together with Hsieh et al., 2018, presents unsupervised object detection & tracking with learnable likelihoods by the means of generative modelling of objects. In particular, our work is the first one to explicitly model object presence, appearance and location through time. Being a generative model, SQAIR can be used for conditional generation, where it can extrapolate sequences into the future. As such, it would be interesting to use it in a reinforcement learning setting in conjunction with Imagination-Augmented Agents (Weber et al., 2017) or more generally as a world model (Ha and Schmidhuber, 2018), especially for settings with simple backgrounds, e. g., games like Montezuma's Revenge or Pacman.

The framework offers various avenues of further research; SQAIR leads to interpretable representations, but the interpretability of *what* variables can be further enhanced by using alternative objectives that disentangle factors of variation in the objects (Kim and Mnih, 2018). Moreover, in its current state, SQAIR can work only with simple backgrounds and static cameras. In future work, we would like to address this shortcoming, as well as speed up the sequential inference process whose complexity is linear in the number of objects. The generative model, which currently assumes additive image composition, can be further improved by e. g., autoregressive modelling (Oord et al., 2016). It can lead to higher fidelity of the model and improved handling of occluded objects. Finally, the SQAIR model is very complex, and it would be useful to perform a series of ablation studies to further investigate the roles of different components.

## Acknowledgements

We would like to thank Ali Eslami for his help in implementing AIR, Alex Bewley and Martin Engelcke for discussions and valuable insights and anonymous reviewers for their constructive feedback. Additionally, we acknowledge that HK and YWT's research leading to these results has received funding from the European Research Council under the European Union's Seventh Framework Programme (FP7/2007-2013) ERC grant agreement no. 617071.

## Footnotes

[2]code: github.com/akosiorek/sqair

[3]video: youtu.be/-IUNQgSLE0c

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
