[Supplementary Material]

## A  Algorithms

Image generation, described by Algorithm 1, is exactly the same for SQAIR and AIR. Algorithms 2 and 3 describe inference in SQAIR. Note that DISC is equivalent to AIR if no latent variables are present in the inputs.

If a function has multiple inputs and if not stated otherwise, all the inputs are concatenated and linearly projected into some fixed-dimensional space, e. g., Lines 9 and 15 in Algorithm 2. Spatial Transformer (ST, e. g., Line 7 in Algorithm 2) has no learnable parameters: it samples a uniform grid of points from an image $\mathbf{x}$, where the grid is transformed according to parameters $\mathbf{z}^{\text{where}}$. $f_{\phi}^1$ is implemented as a perceptron with a single hidden layer. Statistics of $q^P$ and $q^D$ are a result of applying a two-layer multilayer perceptron (MLP) to their respective conditioning sets. Different distributions $q$ do not share parameters of their MLPs. The *glimpse encoder* $h_{\phi}^{\text{glimpse}}$ (Lines 8 and 12 in Algorithm 2 and Line 12 in Algorithm 3; they share parameters) and the *image encoder* $h_{\phi}^{\text{enc}}$ (Line 3 in Algorithm 3) are implemented as two-layer MLPs or convolutional neural networks (CNNs), depending on the experiment (see Appendices D and E for details).

One of the important details of PROP is the proposal glimpse extracted in lines Lines 6 and 7 of Algorithm 2. It has a dual purpose. Firstly, it acts as an information bottleneck in PROP, limiting the flow of information from the current observation $\mathbf{x}_t$ to the updated latent variables $\mathbf{z}_t$. Secondly, even though the information is limited, it can still provide a high-resolution view of the object corresponding to the currently updated latent variable, *given* that the location of the proposal glimpse correctly predicts motion of this object. Initially, our implementation used encoding of the raw observation ($h_{\phi}^{\text{enc}}(\mathbf{x}_t)$, similarly to Line 3 in Algorithm 3) as an input to the relation-RNN (Line 9 in Algorithm 2). We have also experimented with other bottlenecks: (1) low resolution image as an input to the image encoder and (2) a low-dimensional projection of the image encoding before the relation-RNN. Both approaches have led to *ID swaps*, where the order of explaining objects were sometimes swapped for different frames of the sequence (see Figure 10 in Appendix G for an example). Using encoded proposal glimpse extracted from a predicted location has solved this issue.

To condition DISC on propagated latent variables (Line 4 in Algorithm 3), we encode the latter by using a two-layer MLP similarly to Zaheer et al., 2017,

$$\mathbf{l}_t = \sum_{i \in \mathcal{P}_t} \text{MLP}\left(\mathbf{z}_t^{\text{what},i}, \mathbf{z}_t^{\text{where},i}\right). \tag{6}$$

Note that other encoding schemes are possible, though we have experimented only with this one.

---

**Algorithm 1:** Image Generation

**Input**  : $\mathbf{z}_t^{\text{what}}, \mathbf{z}_t^{\text{where}}$ - latent variables from the current time-step.

1  $\mathcal{O}_t = \text{indices}\left(\mathbf{z}_t^{\text{what}}\right)$ `// Indices of all present latent variables.`

2  $\mathbf{y}_t^0 = \mathbf{0}$

3  **for** $i \in \mathcal{O}_t$ **do**

4     $\mathbf{y}_t^{\text{att},i} = f_{\theta}^{\text{dec}}\left(\mathbf{z}_t^{\text{what},i}\right)$ `// Decode the glimpse.`

5     $\mathbf{y}_t^i = \mathbf{y}_t^{i-1} + \text{ST}^{-1}\left(\mathbf{y}_t^{\text{att},i}, \mathbf{z}_t^{\text{where},i}\right)$

6  $\hat{\mathbf{x}}_t \sim \mathcal{N}\left(\boldsymbol{x} \mid \boldsymbol{y}_n, \sigma_x^2 \boldsymbol{I}\right)$

**Output:** $\hat{\boldsymbol{x}}$

---

---

**Algorithm 2:** Inference for Propagation

**Input** : $x_t$ - image at the current time-step,
  $\mathbf{z}_{t-1}^{\text{what}}, \mathbf{z}_{t-1}^{\text{where}}, \mathbf{z}_{t-1}^{\text{pres}}$ - latent variables from the previous time-step
  $\boldsymbol{h}_{t-1}^{T}$ - hidden states from the previous time-step.

1   $\boldsymbol{h}_t^{R,0}, \boldsymbol{z}_t^{\text{what},0}, \boldsymbol{z}_t^{\text{where},0} = \text{initialize}()$
2   $j = 0$ // Index of the object processed in the last iteration.
3   **for** $i \in \mathcal{O}_{t-1}$ **do**
4      **if** $z_{t-1}^{\text{pres},i} == 0$ **then**
5         **continue**
6      $\hat{\mathbf{z}}_t^{\text{where},i} = \text{f}_\phi^1\left(\mathbf{z}_{t-1}^{\text{where},i}, \boldsymbol{h}_t^{T,i}\right)$ // Proposal location.
7      $\hat{\mathbf{g}}_t^i = \text{ST}\left(\mathbf{x}_t, \hat{\mathbf{z}}_t^{\text{where},i}\right)$ // Extract a glimpse from a proposal location.
8      $\hat{\mathbf{e}}_t^i = \text{h}_\phi^{\text{glimpse}}\left(\hat{\mathbf{g}}_t^i\right)$ // Encode the proposal glimpse.
9      $\mathbf{w}_t^{R,i}, \boldsymbol{h}_t^{R,i} = \text{R}_\phi^R\left(\hat{\mathbf{e}}_t^i, \mathbf{z}_{t-1}^{\text{what},i}, \mathbf{z}_{t-1}^{\text{where},i}, \boldsymbol{h}_{t-1}^{T,i}, \boldsymbol{h}_t^{R,j}, \mathbf{z}_t^{\text{what},j}, \mathbf{z}_t^{\text{where},j}\right)$ // Relational
     state, see Equation (14).
10      $\mathbf{z}_t^{\text{where},i} \sim q_\phi^P\left(\mathbf{z}^{\text{where}} \mid \mathbf{z}_{t-1}^{\text{where},k}, \mathbf{w}_t^{R,i}\right)$
11      $\mathbf{g}_t^i = \text{ST}\left(\mathbf{x}_t, \mathbf{z}_t^{\text{where},i}\right)$ // Extract the final glimpse.
12      $\mathbf{e}_t^i = \text{h}_\phi^{\text{glimpse}}\left(\mathbf{g}_t^i\right)$ // Encode the final glimpse.
13      $\mathbf{w}_t^{T,i}, \boldsymbol{h}_t^{T,i} = \text{R}_\phi^T\left(\mathbf{e}_t^i, \mathbf{z}_t^{\text{where},i}, \boldsymbol{h}_{t-1}^{T,i}, \boldsymbol{h}_t^{R,i}\right)$ // Temporal state, see Equation (15).
14      $\mathbf{z}_t^{\text{what},i} \sim q_\phi^P\left(\mathbf{z}^{\text{what}} \mid \mathbf{e}_t^i, \mathbf{z}_{t-1}^{\text{what},i}, \mathbf{w}_t^{R,i}, \mathbf{w}_t^{T,i}\right)$
15      $z_t^{\text{pres},i} \sim q_\phi^P\left(z^{\text{pres}} \mid z_{t-1}^{\text{pres},i}, \mathbf{z}_t^{\text{what},i}, \mathbf{z}_t^{\text{where},i}, \mathbf{w}_t^{R,i}, \mathbf{w}_t^{T,i}\right)$ // Equation (13).
16      $j = i$

**Output** : $\mathbf{z}_t^{\text{what},\mathcal{P}_t}, \mathbf{z}_t^{\text{where},\mathcal{P}_t}, \mathbf{z}_t^{\text{pres},\mathcal{P}_t}$

---

---

**Algorithm 3:** Inference for Discovery

**Input** : $x_t$ - image at the current time-step,
  $\mathbf{z}_t^{\mathcal{P}_t}$ - propagated latent variables for the current time-step,
  $N$ - maximum number of inference steps for discovery.

1   $\boldsymbol{h}_t^{D,0}, \boldsymbol{z}_t^{\text{what},0}, \boldsymbol{z}_t^{\text{where},0} = \text{initialize}()$
2   $j = \text{max\_index}\left(\mathbf{z}_t^{\mathcal{P}_t}\right)$ // Maximum index among the propagated latent variables.
3   $\mathbf{e}_t = \text{h}_\phi^{\text{enc}}\left(\mathbf{x}_t\right)$ // Encode the image.
4   $\mathbf{l}_t = \text{h}_\phi^{\text{enc}}\left(\mathbf{z}_t^{\text{what}}, \mathbf{z}_t^{\text{where}}, \mathbf{z}_t^{\text{pres}}\right)$ // Encode latent variables.
5   **for** $i \in [j+1, \ldots, j+N]$ **do**
6      $\mathbf{w}_t^{D,i}, \boldsymbol{h}_t^{D,i} = \text{R}_\phi^D\left(\mathbf{e}_t, \mathbf{l}_t, \mathbf{z}_t^{\text{what},i-1}, \mathbf{z}_t^{\text{where},i-1}, \boldsymbol{h}_t^{D,i-1}\right)$
7      $z_t^{\text{pres},i} \sim q_\phi^D\left(z^{\text{pres}} \mid \mathbf{w}_t^{D,i}\right)$
8      **if** $z^{\text{pres},i} = 0$ **then**
9         break
10      $\mathbf{z}_t^{\text{where},i} \sim q_\phi^D\left(\mathbf{z}^{\text{where}} \mid \mathbf{w}_t^{D,i}\right)$
11      $\mathbf{g}_t^i = \text{ST}\left(\mathbf{x}_t, \mathbf{z}_t^{\text{where},i}\right)$
12      $\mathbf{e}_t^i = \text{h}_\phi^{\text{glimpse}}\left(\mathbf{g}_t^i\right)$ // Encode the glimpse.
13      $\mathbf{z}_t^{\text{what},i} \sim q_\phi^D\left(\mathbf{z}^{\text{what}} \mid \mathbf{e}_t^i\right)$

**Output** : $\mathbf{z}_t^{\text{what},\mathcal{D}_t}, \mathbf{z}_t^{\text{where},\mathcal{D}_t}, \mathbf{z}_t^{\text{pres},\mathcal{D}_t}$

---

## B    Details for the Generative Model of SQAIR

In implementation, we upper bound the number of objects at any given time by $N$. In detail, the discovery prior is given by

$$p^D\Big(D_t, \mathbf{z}_t^{\mathcal{D}_t} \mid \mathbf{z}_t^{\mathcal{P}_t}\Big) = p^D(D_t \mid P_t) \prod_{i \in \mathcal{D}_t} p^D(\mathbf{z}_t^{\text{what},i}) p^D(\mathbf{z}_t^{\text{where},i}) \delta_1(z_t^{\text{pres},i}), \qquad (7)$$

$$p^D(D_t \mid P_t) = \text{Categorical}\left(D_t; N - P_t, p_\theta(P_t)\right), \qquad (8)$$

where $\delta_x(\cdot)$ is the delta function at $x$, Categorical$(k; K, p)$ implies $k \in \{0, 1, \ldots, K\}$ with probabilities $p_0, p_1, \ldots, p_K$ and $p^D(\mathbf{z}_t^{\text{what},i}), p^D(\mathbf{z}_t^{\text{where},i})$ are fixed isotropic Gaussians. The propagation prior is given by

$$p^P\Big(\mathbf{z}_t^{\mathcal{P}_t} \mid \mathbf{z}_{t-1}\Big) = \prod_{i \in \mathcal{P}_t} p^P\Big(\mathbf{z}_t^{\text{pres},i} \mid \mathbf{z}_{t-1}^{\text{pres},i}, \boldsymbol{h}_{t-1}\Big) p^P\Big(\mathbf{z}_t^{\text{what},i} \mid \boldsymbol{h}_{t-1}\Big) p^P\Big(\mathbf{z}_t^{\text{where},i} \mid \boldsymbol{h}_{t-1}\Big), \quad (9)$$

$$p^P\Big(\mathbf{z}_t^{\text{pres},i} \mid \mathbf{z}_{t-1}^{\text{pres},i}, \boldsymbol{h}_{t-1}\Big) = \text{Bernoulli}(z_t^{\text{pres},i}; f_\theta(\boldsymbol{h}_{t-1})) \delta_1(z_{t-1}^{\text{pres},i}), \qquad (10)$$

with $f_\theta$ a scalar-valued function with range $[0, 1]$ and $p^P(\mathbf{z}_t^{\text{what},i}|\boldsymbol{h}_{t-1})$, $p^P(\mathbf{z}_t^{\text{where},i}|\boldsymbol{h}_{t-1})$ both factorised Gaussians parameterised by some function of $\boldsymbol{h}_{t-1}$.

## C    Details for the Inference of SQAIR

The propagation inference network $q_\phi^P$ is given as below,

$$q_\phi^P\Big(\mathbf{z}_t^{\mathcal{P}_t} \mid \mathbf{x}_t, \mathbf{z}_{t-1}, \boldsymbol{h}_t^{T,\mathcal{P}_t}\Big) = \prod_{i \in \mathcal{O}_{t-1}} q_\phi^P\Big(\mathbf{z}_t^i \mid \mathbf{x}_t, \mathbf{z}_{t-1}^i, \boldsymbol{h}_t^{T,i}, \boldsymbol{h}_t^{R,i}\Big), \qquad (11)$$

with $\boldsymbol{h}_t^{R,i}$ the hidden state of the relation RNN (see Equation (14)). Its role is to capture information from the observation $\mathbf{x}_t$ as well as to model dependencies between different objects. The propagation posterior for a single object can be expanded as follows,

$$
\begin{aligned}
q_\phi^P & \Big(\mathbf{z}_t^i \mid \mathbf{x}_t, \mathbf{z}_{t-1}^i, \boldsymbol{h}_t^{T,i}, \boldsymbol{h}_t^{R,i}\Big) = \\
& q_\phi^P\Big(\mathbf{z}_t^{\text{where},i} \mid \mathbf{z}_{t-1}^{\text{what},i}, \mathbf{z}_{t-1}^{\text{where},i}, \boldsymbol{h}_{t-1}^{T,i}, \boldsymbol{h}_t^{R,i}\Big) \\
& q_\phi^P\Big(\mathbf{z}_t^{\text{what},i} \mid \mathbf{x}_t, \mathbf{z}_t^{\text{where},i}, \mathbf{z}_{t-1}^{\text{what},i}, \boldsymbol{h}_t^{T,i}, \boldsymbol{h}_t^{R,i}\Big) \\
& q_\phi^P\Big(z_t^{\text{pres},i} \mid \mathbf{z}_t^{\text{what},i}, \mathbf{z}_t^{\text{where},i}, z_{t-1}^{\text{pres},i}, \boldsymbol{h}_t^{T,i}, \boldsymbol{h}_t^{R,i}\Big).
\end{aligned}
\qquad (12)
$$

In the second line, we condition the object location $\mathbf{z}_t^{\text{where},i}$ on its previous appearance and location as well as its dynamics and relation with other objects. In the third line, current appearance $\mathbf{z}_t^{\text{what},i}$ is conditioned on the new location. Both $\mathbf{z}_t^{\text{where},i}$ and $\mathbf{z}_t^{\text{what},i}$ are modelled as factorised Gaussians. Finally, presence depends on the new appearance and location as well as the presence of the same object at the previous time-step. More specifically,

$$
\begin{aligned}
q_\phi^P & \Big(z_t^{\text{pres},i} \mid \mathbf{z}_t^{\text{what},i}, \mathbf{z}_t^{\text{where},i}, z_{t-1}^{\text{pres},i}, \boldsymbol{h}_t^{T,i}, \boldsymbol{h}_t^{R,i}\Big) \\
& = \text{Bernoulli}\Big(z_t^{\text{pres},i} \mid f_\phi\Big(\mathbf{z}_t^{\text{what},i}, \mathbf{z}_t^{\text{where},i}, \boldsymbol{h}_t^{T,i}, \boldsymbol{h}_t^{R,i}\Big)\Big) \delta_1(z_{t-1}^{\text{pres},i}),
\end{aligned}
\qquad (13)
$$

where the second term is the delta distribution centered on the presence of this object at the previous time-step. If it was not there, it cannot be propagated. Let $j \in \{0, \ldots, i-1\}$ be the index of the most recent present object before object $i$. Hidden states are updated as follows,

$$\boldsymbol{h}_t^{R,i} = \text{R}_\phi^R\Big(\mathbf{x}_t, \mathbf{z}_{t-1}^{\text{what},i}, \mathbf{z}_{t-1}^{\text{where},i}, \boldsymbol{h}_{t-1}^{T,i}, \boldsymbol{h}_t^{R,i-1}, \mathbf{z}_t^{\text{what},j}, \mathbf{z}_t^{\text{where},j}\Big), \qquad (14)$$

$$\boldsymbol{h}_t^{T,i} = \text{R}_\phi^T\Big(\mathbf{x}_t, \mathbf{z}_t^{\text{where},i}, \boldsymbol{h}_{t-1}^{T,i}, \boldsymbol{h}_t^{R,i}\Big), \qquad (15)$$

where $\text{R}_\phi^T$ and $\text{R}_\phi^R$ are temporal and propagation RNNs, respectively. Note that in Eq. (14) the RNN does not have direct access to the image $\mathbf{x}_t$, but rather accesses it by extracting an attention glimpse at a proposal location, predicted from $\boldsymbol{h}_{t-1}^{T,i}$ and $\mathbf{z}_{t-1}^{\text{where},i}$. This might seem like a minor detail, but in practice structuring computation this way prevents ID swaps from occurring, *cf.* Appendix G. For computational details, please see Algorithms 2 and 3 in Appendix A.

# D   Details of the moving-MNIST Experiments

## D.1   SQAIR and AIR Training Details

All models are trained by maximising the evidence lower bound (ELBO) $\mathcal{L}_{IWAE}$ (Equation (5)) with the RMSPROP optimizer (Tieleman and Hinton, 2012) with momentum equal to $0.9$. We use the learning rate of $10^{-5}$ and decrease it to $\frac{1}{3} \cdot 10^{-5}$ after 400k and to $10^{-6}$ after 1000k training iterations. Models are trained for the maximum of $2 \cdot 10^6$ training iterations; we apply early stopping in case of overfitting. SQAIR models are trained with a curriculum of sequences of increasing length: we start with three time-steps, and increase by one time-step every $10^5$ training steps until reaching the maximum length of 10. When training AIR, we treated all time-steps of a sequence as independent, and we trained it on all data (sequences of length ten, split into ten independent sequences of length one).

## D.2   SQAIR and AIR Model Architectures

All models use glimpse size of $20 \times 20$ and exponential linear unit (ELU) (Clevert et al., 2015) non-linearities for all layers except RNNs and output layers. MLP-SQAIR uses fully-connected layers for all networks. In both variants of SQAIR, the $\mathrm{R}_\phi^D$ and $\mathrm{R}_\phi^R$ RNNs are the vanilla RNNs. The propagation prior RNN and the temporal RNN $\mathrm{R}_\phi^T$ use gated recurrent unit (GRU). AIR follows the same architecture as MLP-SQAIR. All fully-connected layers and RNNs in MLP-SQAIR and AIR have 256 units; they have 2.9M and 1.7M trainable parameters, respectively.

CONV-SQAIR differs from the MLP version in that it uses CNNs for the glimpse and image encoders and a subpixel-CNN (Shi et al., 2016) for the glimpse decoder. All fully connected layers and RNNs have 128 units. The encoders share the CNN, which is followed by a single fully-connected layer (different for each encoder). The CNN has four convolutional layers with $[16, 32, 32, 64]$ features maps and strides of $[2, 2, 1, 1]$. The glimpse decoder is composed of two fully-connected layers with $[256, 800]$ hidden units, whose outputs are reshaped into 32 features maps of size $5 \times 5$, followed by a subpixel-CNN with three layers of $[32, 64, 64]$ feature maps and strides of $[1, 2, 2]$. All filters are of size $3 \times 3$. CONV-SQAIR has 2.6M trainable parameters.

We have experimented with different sizes of fully-connected layers and RNNs; we kept the size of all layers the same and altered it in increments of 32 units. Values greater than 256 for MLP-SQAIR and 128 for CONV-SQAIR resulted in overfitting. Models with as few as 32 units per layer ($< 0.9$M trainable parameters for MLP-SQAIR) displayed the same qualitative behaviour as reported models, but showed lower quantitative performance.

The output likelihood used in both SQAIR and AIR is Gaussian with a fixed standard deviation set to $0.3$, as used by Eslami et al., 2016. We tried using a learnable scalar standard deviation, but decided not to report it due to unsable behaviour in the early stages of training. Typically, standard deviation would converge to a low value early in training, which leads to high penalties for reconstruction mistakes. In this regime, it is beneficial for the model to perform no inference steps ($z^{\mathrm{pres}}$ is always equal to zero), and the model never learns. Fixing standard deviation for the first 10k iterations and then learning it solves this issue, but it introduces unnecessary complexity into the training procedure.

## D.3   VRNN Implementation and Training Details

Our VRNN implementation is based on the implementation[4] of Filtering Variational Objectives (FIVO) by Maddison et al., 2017. We use an LSTM with hidden size $J$ for the deterministic backbone of the VRNN. At time $t$, the LSTM receives $\psi^x(\mathbf{x}_{t-1})$ and $\psi^z(\mathbf{z}_{t-1})$ as input and outputs $o_t$, where $\psi^x$ is a data feature extractor and $\psi^z$ is a latent feature extractor. The output is mapped to the mean and standard deviation of the Gaussian prior $p_\theta(\mathbf{z}_t \mid \mathbf{x}_{t-1})$ by an MLP. The likelihood $p_\theta(\mathbf{x}_t \mid \mathbf{z}_t, \mathbf{x}_{t-1})$ is a Gaussian, with mean given by $\psi^{\mathrm{dec}}(\psi^z(\mathbf{z}_t), o_t)$ and standard deviation fixed to be $0.3$ as for SQAIR and AIR. The inference network $q_\phi(\mathbf{z}_t \mid \mathbf{z}_{t-1}, \mathbf{x}_t)$ is a Gaussian with mean and standard deviation given by the output of separate MLPs with inputs $[o_t, \psi^x(\mathbf{x}_t)]$.

All aforementioned MLPs use the same number of hidden units $H$ and the same number of hidden layers $L$. The CONV-VRNN uses a CNN for $\psi^x$ and a transposed CNN for $\psi^{\mathrm{dec}}$. The MLP-VRNN uses an MLP with $H'$ hidden units and $L'$ hidden layers for both. ELU were used throughout as activations. The latent dimensionality was fixed to 165, which is the upper bound of the number of

Table 2: Number of trainable parameters for the reported models.

| | CONV-SQAIR | MLP-SQAIR | MLP-AIR | CONV-VRNN | MLP-VRNN |
|---|---|---|---|---|---|
| number of parameters | 2.6M | 2.9M | 1.7M | 2.6M | 2.1M |

latent dimensions that can be used per time-step in SQAIR or AIR. Training was done by optimising the FIVO bound, which is known to be tighter than the IWAE bound for sequential latent variable models (Maddison et al., 2017). We also verified that this was the case with our models on the moving-MNIST data. We train with the RMSPROP optimizer with a learning rate of $10^{-5}$, momentum equal to $0.9$, and training until convergence of test FIVO bound.

For each of MLP-VRNN and CONV-VRNN, we experimented with three architectures: small/medium/large. We used $H=H'=J$=128/256/512 and $L=L'$=2/3/4 for MLP-VRNN, giving number of parameters of 1.2M/2.1M/9.8M. For CONV-VRNN, the number of features maps we used was $[32, 32, 64, 64]$, $[32, 32, 32, 64, 64, 64]$ and $[32, 32, 32, 64, 64, 64, 64, 64, 64]$, with strides of $[2, 2, 2, 2]$, $[1, 2, 1, 2, 1, 2]$ and $[1, 2, 1, 2, 1, 2, 1, 1, 1]$, all with $3 \times 3$ filters, $H=J$=128/256/512 and $L$=1, giving number of parameters of 0.8M/2.6M/6.1M. The largest convolutional encoder architecture is very similar to that in Gulrajani et al., 2016 applied to MNIST.

We have chosen the medium-sized models for comparison with SQAIR due to overfitting encountered in larger models.

### D.4 Addition Experiment

We perform the addition experiment by feeding latent representations extracted from the considered models into a 19-way classifier, as there are 19 possible outputs (addition of two digits between 0 and 9). The classifier is implemented as an MLP with two hidden layers with 256 ELU units each and a softmax output. For AIR and SQAIR, we use concatenated $\mathbf{z}^{\mathrm{what}}$ variables multiplied by the corresponding $z^{\mathrm{pres}}$ variables, while for VRNN we use the whole 165-dimensional latent vector. We train the model over $10^7$ training iterations with the ADAM optimizer (Kingma and Ba, 2015) with default parameters (in tensorflow).

## E   Details of the *DukeMTMC* Experiments

We take videos from cameras one, two, five, six and eight from the *DukeMTMC* dataset (Ristani et al., 2016). As pre-processing, we invert colors and subtract backgrounds using standard OpenCV tools (Itseez, 2015), downsample to the resolution of $240 \times 175$, convert to gray-scale and randomly crop fragments of size $64 \times 64$. Finally, we generate 3500 sequences of length five such that the maximum number of objects present in any single frame is three and we split them into training and validation sets with the ratio of $9 : 1$.

We use the same training procedure as for the MNIST experiments. The only exception is the learning curriculum, which goes from three to five time-steps, since this is the maximum length of the sequences.

The reported model is similar to CONV-SQAIR. We set the glimpse size to $28 \times 12$ to account for the expected aspect ratio of pedestrians. Glimpse and image encoders share a CNN with $[16, 32, 64, 64]$ feature maps and strides of $[2, 2, 2, 1]$ followed by a fully-connected layer (different for each encoder). The glimpse decoder is implemented as a two-layer fully-connected network with 128 and 1344 units, whose outputs are reshaped into 64 feature maps of size $7 \times 3$, followed by a subpixel-CNN with two layers of $[64, 64]$ feature maps and strides of $[2, 2]$. All remaining fully-connected layers in the model have 128 units. The total number of trainable parameters is 3.5M.

## F   Harder multi-MNIST Experiment

We created a version of the multi-MNIST dataset, where objects can appear or disappear at an arbitrary point in time. It differs from the dataset described in Section 4.1, where all digits are present throughout the sequence. All other dataset parameters are the same as in Section 4.1. Figure 9 shows an example sequence and MLP-SQAIR reconstructions with marked glimpse locations. The model has no trouble detecting new digits in the middle of the sequence and rediscovering a digit that was previously present.

Figure 9: SQAIR trained on a harder version of moving-MNIST. Input images (top) and SQAIR reconstructions with marked glimpse locations (bottom)

# G   Failure cases of SQAIR

Figure 10: Examples of ID swaps in a version of SQAIR *without* proposal glimpse extraction in PROP (see Appendix A for details). Bounding box colours correspond to object index (or its identity). When PROP is allowed the same access to the image as DISC, then it often prefers to ignore latent variables, which leads to swapped inference order.

Figure 11: Examples of re-detections in MLP-SQAIR. Bounding box colours correspond to object identity, assigned to it upon discovery. In some training runs, SQAIR converges to a solution, where objects are re-detected in the second frame, and PROP starts tracking only in the third frame (left). Occasionally, an object can be re-detected after it has severely overlapped with another one (top right). Sometimes the model decides to use only DISC and repeatedly discovers all objects (bottom right). These failure mode seem to be mutually exclusive – they come from different training runs.

Figure 12: Two failed reconstructions of SQAIR. *Left*: SQAIR re-detects objects in the second time-step. Instead of 5 and 2, however, it reconstructs them as 6 and 7. Interestingly, reconstructions are consistent through the rest of the sequence. *Right:* At the second time-step, overlapping 6 and 8 are explained as 6 and a small 0. The model realizes its mistake in the third time-step, re-detects both digits and reconstructs them properly.

# H    Reconstruction and Samples from the Moving-MNIST Dataset

## H.1    Reconstructions

Figure 13: Sequences of input (first row) and SQAIR reconstructions with marked glimpse locations. Reconstructions are all temporally consistent.

Figure 14: Sequences of input (first row) and CONV-VRNN reconstructions. They are not temporally consistent. The reconstruction at time $t = 1$ is typically of lower quality and often different than the rest of the sequence.

## H.2 Samples

Figure 15: Samples from SQAIR. Both motion and appearance are temporally consistent. In the last sample, the model introduces the third object despite the fact that it has seen only up to two objects in training.

Figure 16: Samples from CONV-VRNN. They show lack of temporal consistency. Objects in the generated frames change between consecutive time-steps and they do not resamble digits from the training set.

## H.3 Conditional Generation

Figure 17: Conditional generation from SQAIR, which sees only the first three frames in every case. Top is the input sequence (and the remaining ground-truth), while bottom is reconstruction (first three time-steps) and then generation.

# I  Reconstruction and Samples from the DukeMTMC Dataset

Figure 18: Sequences of input (first row) and SQAIR reconstructions with marked glimpse locations. While not perfect (spurious detections, missed objects), they are temporally consistent and similar in appearance to the inputs.

Figure 19: Samples with marked glimpse locations from SQAIR trained on the DukeMTMC dataset. Both appearance and motion is spatially consistent. Generated objects are similar in appearance to pedestrians in the training data. Samples are noisy, but so is the dataset.

Figure 20: Conditional generation from SQAIR, which sees only the first four frames in every case. Top is the input sequence (and the remaining ground-truth), while bottom is reconstruction (first four time-steps) and then generation.

## Footnotes

[4]`https://github.com/tensorflow/models/tree/master/research/fivo`