[Reviews · NeurIPS 2018]

Reviewer 1



This paper improves the existing Attend, Infer, Repeat paper for sequential data and moving objects. This is a much needed improvement of the AIR technique and the results are very promising. I have a few questions to the others and suggestions of improvements: 1,The sequence length is fixed and predefined. For multi-MNIST, the length is 10 and thus the model is always trained on models of length 10. What if the data has variable lengths? What if the SQAIR model is trained on different sequence lengths - make the length as a hyperparameter and show the performance. 2. A more practical experimentation would be on any self driving car dataset. It would be great if the authors could add results on that. 3. The details of the models and the analysis results are added in the appendix and in supplementary. Its an important part of the paper and should be included in the paper.

Reviewer 2



I have read the other reviews and the author rebuttal. I am still very much in favor of accepting this paper, but I have revised my score down from a 9 to an 8; some of the issues pointed out by the other reviewers, while well-addressed in the rebuttal, made me realize that my initial view of the paper was a bit too rosy. ------------------------ This paper presents a deep generative model for unsupervised modeling of moving objects in image sequences. The model starts with the basic Attend, Infer, Repeat (AIR) framework and extends it to handle images sequences (SQAIR). This extension requires taking into account the fact that objects may enter into or leave the frame over the course of a motion sequence. To support this behavior, SQAIR's generative and inference networks for each frame have two phases. First, a *propagation* network extrapolates the positions of existing objects forward in time, then decides whether any of those objects are no longer in the frame and should be 'forgotten.' Second, a *discovery* network proposes new objects that have entered the frame, conditioned on the output of the propagation network. SQAIR is quantitatively evaluated on a moving MNIST dataset by its test-set NLL, image reconstruction accuracy, divergence between learned prior and approximate posterior, accuracy at counting the number of objects, and accuracy on a supervised task involving predicting the sums of digits present in the image using the learned latent representation. It is also qualitatively evaluated for its ability to reconstruct, complete, and generate image sequences both on the moving MNIST dataset and on the DukeMTMC pedestrain CCTV dataset. SQAIR outperforms AIR and VRNN, and is qualitatively better able to handle occlusions and objects entering/leaving the frame. This is very good work. Even a few years ago, a model that can perform unsupervised object detection and tracking in video sequences would have been unthinkable--this work points toward a future where that is very possible. Now, given the existence of AIR, perhaps a sequential extension of AIR might seem like an obvious idea. However, getting the implementation of that idea right is far from trivial, and the architecture proposed here (propagation + discovery) is well-motivated and seems to work well. The paper is written clearly and provides sufficient detail and evaluation. I am very much in favor of accepting this paper.

Reviewer 3



The authors propose a temporal extension of the structured image generative model AIR. The model is composed of two parts: the first one, DISC, detects new objects at each time step, and is very similar to AIR. The only difference is that its predictions are conditioned on the updated latent variables of previous frames. The second one PROP, is responsable for updating or forgetting latent variables given a new image. It does so by employing two RNNs : the first one (temporal) updates the previous latent states. The second one (relational) models relations between objects, and is recurrent over the objects in the frame. The authors show extensive experimentation on a stochastic moving MNIST dataset and qualitative results on a dataset collected from static CCTV cameras. The structure of their model allows them to learn to detect and track objects in an unsupervised fashion, perform video prediction, and use the latent variables for solving simple tasks (addition, counting of objects). The authors propose a compelling and ambitious approach that tackles the extremely important problem of structured generative modeling, where the structure can be used for unsupervised learning of high level recognition tasks. Incorporating structure in the model to reflect the prior that a scene is composed of objects that should be mainly described by their location and their appearance, and showing that the learned latent variables are interpretable and thus useful in down stream tasks, is an extremely appealing direction. At this point, the approach does not model the background, and has not been shown to be applicable in contexts where the camera is non static (as background subtraction is necessary), but it constitues a nice step towards in this important direction. It is nice that the authors show a real-world application of their method. Unfortunately, the experimental evaluation is rather lacking, and I have several questions in regards to this matter: - the authors should compare to other VAE-flavoured state of the art video generative modelling methods, such as Denton et al. ICML 2018. In particular, the qualitative results of Figure 12 are clearly not state of the art. - how important is the relational RNN ? The authors transparently acknowledge that the method is complex and an ablation study would be very useful. It would also be helpful to show in a more tangible manner how this RNN is "implementing the explaining away phenomenon" ? l.122 - "All objects are present through the sequence; the digits can move out of the frame, but always come back." l.152, 153 Why impose this constraint on the dataset ? What severely limits the model from dealing where cases when the digit does not come back ? Also, can the digits appear during the sequence and if not, why not ? If this showed failures of the model, it would be interesting to analyse why. - I don't agree with the justifications for not filling in Table 1: a classifier could be learned for VRNN for counting, just like in the addition experiment. And for completion, why not perform these experiments for the MLP-VRNN ; it is better in terms of KL divergence than its CONV counterpart. - "We will show that the resulting model, which we name Sequential AIR (Sequential Attend, Infer, Repeat (SQAIR )), retains the strengths of the original AIR formulation while outperforming it in both synthetic and real-world scenarios." I did not find the results for AIR on the CCTV dataset. - can you give a qualitative interpretation of the fact that the reconstruction metric of MLP-AIR is better than MLP-SQAIR ? - Figure 10: why are there two entirely lines input fully black inputs ? - "which signifies that the latent representation can be allocated to regions of the frames that need it, as opposed to describing the whole scene" l.170 This is not clear to me. Cumulatively, this leaves the reader a little bit unsatisfied (to summarize: missing results, no ablation study, missing comparison to an important state of the art paper, main results on a toyish dataset that could be slightly less toy). Also, it would strengthen the paper a lot to show a more thorough and quantitative evaluation on the real-world dataset. This is what leads me to the current decision of "Marginally below the acceptance threshold." Regarding clarity of the method, the paper is generally well written, but I would like to have the following clarifications: - l. 118 you specify that the temporal RNN has no access to the image. Then in appendix D, equation (14), as well as Figure 3, imply the opposite. Can you clarify ? - Figure 6 is ambiguous: the lower lines are predictions or reconstructions ? Finally, a note that for this type of problem, accompanying videos are much appreciated to ease the qualitative evaluation of the samples and reconstructions, especially to appreciate temporal consistency. Typos / errors : - then should be than l.53 and l.188 - l.59 in the AIR model, n is at most N (a fixed hyperparameter of the method) - l.65: according to Figure 1, f_theta^dec 's input is z_t^what,i and output is g_t^i. Calling the decoder a "glimpse decoder" seems to imply some form of attention mechanism on the generated image, which is not the case. Also, z^where should be indexed with i. - l.97 vector valued should be -valued vector - l.103 logical flow of text would require that it is also explained that z^what and z^where are sampled during the propagation step - images l.151 should be digits - l.160 Being similar to a VAE. -------- Assessment after the rebuttal -------- The authors have addressed most of my concerns. The experimental protocol is still slightly lacking, as they do not experimentally validate the impact of using a relational RNN on top of the temporal RNN; and I still think that the toy dataset should have had certain properties to test the strengths of the proposed model (like appearing and disappearing digits.) However, like the authors and the other reviewers, I am convinced of the importance of the problem addressed, and I acknowledge that the authors have proposed a compelling and ambitious approach to tackle this problem. Therefore, with the clarifications and improvements that the authors have promised to bring, I now think that this paper is good and should be accepted.